# Guideline Adherence and Registry Recruitment of Congenital Primary Hypothyroidism: Data from the German Registry for Congenital Hypothyroidism (HypoDok)

**DOI:** 10.3390/ijns7010010

**Published:** 2021-02-12

**Authors:** Julia Thomann, Sascha R. Tittel, Egbert Voss, Rudolf Oeverink, Katja Palm, Susanne Fricke-Otto, Klaus Kapelari, Reinhard W. Holl, Joachim Woelfle, Markus Bettendorf

**Affiliations:** 1Division of Paediatric Endocrinology and Diabetes, University Children’s Hospital Heidelberg, 69120 Heidelberg, Germany; markus.bettendorf@med.uni-heidelberg.de; 2Institute of Epidemiology and Medical Biometry, Central Institute for Biomedical Technology, Ulm University, 89081 Ulm, Germany; sascha.tittel@uni-ulm.de (S.R.T.); reinhard.holl@uni-ulm.de (R.W.H.); 3Cnopfsche Kinderklinik, 90419 Nürnberg, Germany; egbert.voss@diakoneo.de; 4Medicover Paediatric Endocrinology, 26122 Oldenburg, Germany; rudolf.oeverink@medicover.de; 5Department of Paediatrics, University Hospital Magdeburg, 39120 Magdeburg, Germany; Katja.Palm@med.ovgu.de; 6Children’s Hospital Krefeld, 47805 Krefeld, Germany; susanne.fricke-otto@helios-gesundheit.de; 7Division of Paediatric Endocrinology and Diabetes, University Hospital Innsbruck, 6020 Innsbruck, Austria; klaus.kapelari@i-med.ac.at; 8Department of Paediatrics, University Hospital Erlangen, 91054 Erlangen, Germany; Joachim.Woelfle@uk-erlangen.de

**Keywords:** standardized documentation, quality assurance, longitudinal comparison, multicenter database

## Abstract

Neonatal screening for congenital primary hypothyroidism (CH) is mandatory in Germany but medical care thereafter remains inconsistent. Therefore, the registry HypoDok of the German Society of Pediatric Endocrinology and Diabetology (DGKED) was analyzed to evaluate the implementation of evidence-based guidelines and to assess the number of included patients. Inclusion criteria were (i) date of birth between 10/2001 and 05/2020 and (ii) increased thyroid-stimulating hormone (TSH) at screening and/or confirmation. The cohort was divided into before (A) and after (B) guideline publication in 02/2011, to assess the guideline’s influence on medical care. A total of 659 patients were analyzed as group A (*n* = 327) and group B (*n* = 332) representing 17.5% and 10.3% of CH patients identified in the German and Austrian neonatal screening program during the respective time period. Treatment start and thyroxine doses were similar in both groups and consistent with recommendations. Regular follow-ups were documented. In the first three years of life, less than half of the patients underwent audiometry; developmental assessment was performed in 49.3% (A) and 24.8% (B) (*p* < 0.01). Documentation of CH patient care by pediatric endocrinologists seemed to be established, however, it reflected only a minority of the affected patients. Therefore, comprehensive documentation as an important instrument of quality assurance and evidence-based medicine should be legally enforced and officially funded in order to record, comprehend, and optimize care and outcome in patients with rare diseases such as CH.

## 1. Introduction

Congenital primary hypothyroidism (CH) as a rare disease is the most common endocrine disease with an overall prevalence of 1:2000–3000 in Europe. CH prevalence shows regional heterogeneity within Europe with recent studies reporting increasing higher incidences between 1:1000 and 1:1500 in Macedonia and parts of Italy [1,2]. Between 2004 and 2017, 160 to 280 CH patients were detected in the neonatal screening in Germany per year [3].

After introduction of the neonatal screening in Germany and Austria in the 1970s, the symptoms of CH such as growth retardation and delay of psychomotor development could be prevented by an early treatment start with levothyroxine (L-T_4_) [4,5].

Guidelines were introduced to harmonise CH treatment strategies and allow evidence-based patient care [6]. Circulation and compulsory implementation of medical guidelines in clinical care remain limited [7]. Adherence to clinical guidelines varies considerably. Reinforcement and education are essential to enhance overall adherence [8].

The German guideline for CH [9] was published in 02/2011 in accordance with international recommendations and is currently under revision [10,11]. It is the first German guideline for CH considering the available evidence up to date. The core statements reflect evidence level S2k (consensus-based guideline) are highlighted in Table 1 and were evaluated with the HypoDok registry. The missing three statements, which aren’t evaluated in this manuscript, can be found in Appendix A. The Austrian Working Group for Pediatric Endocrinology and Diabetology (APEDÖ) revised its recommendations based on the German guideline in 2017 [12], which is merely a supplement to the existing German guideline.

Registries are built to improve patient care and to install instruments of quality control in treating rare endocrine diseases [13]. They also collect data for epidemiological and clinical research activities. The German Society of Pediatric Endocrinology and Diabetology (DGKED) launched the registry HypoDok as an initiative for quality improvement [14,15]. The aim of this study was to compare the documented care for patients with CH to the evidence-based guideline recommendations in order to explore guideline adherence and to assess overall acceptance of HypoDok as a tool for quality assessment.

## 2. Patients/Methods

Based on the anonymized multicenter data collected from the registry, a descriptive evaluation was carried out as an observational study. The data were collected prospectively using the software HypoDok, which was developed at the Institute for Epidemiology and Medical Biometry at Ulm University and distributed free of charge to all interested centers in Germany and Austria [15].

The registry was initiated by the German Society for Pediatric Endocrinology and Diabetology (DGKED) on 01/2000 to harmonize the quality of diagnostics and treatment of endocrine disorders across Germany and Austria. For the transfer of pseudonymized data for analysis, consent was obtained from the State Data Protection Commissioner for Saxony-Anhalt. Data collection has been approved centrally by the ethics committee of the University of Magdeburg.

Patient selection and registration were at the discretion of the treating physicians. Patient data was manually extracted from the patient chart and individually entered in a decentralized manner in the local database. Twice a year the local pseudonymized patient data was extracted, exported, and electronically transmitted to the registry in Ulm.

Fifty participating centers in Germany (48) and Austria (2) recorded data of a total of 1630 patients (as of 05/2020) with CH in HypoDok. Inclusion criteria for this analysis were (i) date of birth between 10/2001 and 05/2020 and (ii) elevated screening TSH and/or elevated serum TSH at confirmation (>10 mU/l) (Figure 1). An elevated screening TSH (whole blood) was defined in the database when the option “Screening TSH normal: no” was ticked or when values >15 mU/l (cut-off) were entered. To gain comparable populations for data analysis nine years after the introduction of the guideline (2011), the period prior to the introduction of the guideline was equally selected as nine years. As a result, a total of 659 HypoDok patients fulfilled the inclusion criteria. For the other patients in the registry, either the date of birth was outside the defined period or the data entry was incomplete/inclusion criteria (ii) were missing.

The primary target parameters were defined based on the statements of the German guideline [16]: age at start of therapy [days of life]; serum TSH [mU/L]; serum T4 [µg/dL]; serum fT4 [ng/L]; recommended L-T_4_ dose [µg/day]; duration until TSH normalization (<10 µU/mL) after start of therapy [days]; developmental assessment result normal [yes/no]; presentation in 1/2/4 weeks after start of therapy [yes/no]; audiometry [yes/no]; thyroid ultrasound [yes/no]; thyroid scintigraphy [yes/no]; thyroid peroxidase antibody (TPO-Ab)/thyroid-stimulating hormone receptor antibody (TSHR-Ab) determined [yes/no]. A normal developmental assessment was defined by the previous assessment of the treatment facility: developmental assessment result normal: yes. The parameters presented in the Appendix A were defined as following: thyroid gland in loco typico [yes/no]; athyreosis [yes/no]; ectopia [yes/no]. Missing values were excluded from analysis.

The target parameters were evaluated separately in a descriptive approach using SAS Version 9.4. The results are presented as median and interquartile range (IQR; continuous variables) or as percent (binary and categorical variables). The differences between the two groups are calculated for the continuous parameters using the Wilcoxon rank sum test and for the percentages using the Chi-square test. The *p*-values were adjusted for multiple testing using the Holm method. A two-sided *p*-value of less than 0.05 was defined to be significant.

To assess the coverage of the registry, the number of patients recorded in HypoDok and included in this study was compared with the number of new diagnoses of congenital hypothyroidism detected in the German/Austrian newborn screening per year. Data of the German neonatal screening report for the years 2004–2017 and data from Austria for the years 2015–2019 were available. From the German newborn screening report, the positive predictive value was available for the years 2004–2011 and 2017, specificity was available for the years 2005–2009 and 2017.

## 3. Results

Demographic characteristics of all 659 patients (group A: *n* = 327 patients; B: *n* = 332 patients) are listed in Table 2. Of these patients, 2.2% (7 in total) in group A and 6% (20 in total) in group B were Austrian; 63.6% and 60.5% of the patients in groups A and B, respectively, were female.

A comparison of patients in the registry with those from the German neonatal screening registry showed that HypoDok documented an average of 17.5% of the patients registered by the German neonatal screening with confirmed diagnosis in the years 2004–2017. Before the introduction of the guideline 19.1% of the patients were documented; after the introduction of the guideline 15.8% of the patients were documented. In the latest publication from 2017, the prevalence of CH was 1:2813. Recall rate (0.13%), positive predictive value (PPV) (28.01%), and specificity (99.91%) confirm high quality of CH screening. All quality measures obtained in the respective years of analysis of the German newborn screening are provided in Appendix A.

Austrian data were only available after the introduction of the guideline; here the documentation rate in the HypoDok database was 10.3% compared to the Austrian neonatal screening registry (Table 3).

For most of the target parameters, not all patients had data available, so the number of evaluated patient data is shown separately:

The complete evaluation of core statements of the guideline with regard to diagnostics is listed in Table 4a. The confirmation of the CH diagnosis by determination of fT4 and/or TT4 was documented in 34.9% (*n* = 327) of the patients in group A and in 24.4% (*n* = 332) of the patients in group B (*p* = 0.02). Audiometry was performed in 24.4% of patients in group A (*n* = 172) in the time period 14 days before/after the start of therapy with L-T_4_, and in group B in 11.8% (*n* = 229) within this time interval (*p* = 0.01). Subgrouping of patients based on ultrasonography are shown in the Appendix A. Among the patients available for evaluation in this setting, 7.4% (group A, *n* = 54) and 6.7% (group B, *n* = 90) had an ectopic located thyroid gland, 24.1% (group A, *n* = 58) and 38.5% (group B, *n* = 96) showed athyreosis. Gene mutations indicating dyshormonogenesis (mutations in SLC26A4, thyroid peroxidase, and thyreoglobulin gene) are found more frequently compared to genetic alterations associated with dysgenesis (mutation in NKx2.1/TTF1, PAX8, and TSH-receptor gene) in both groups (Appendix A).

With no significant differences between the groups initially, therapy was carried out according to the guidelines as shown in Table 4b.

Table 4c shows the evaluation of the guideline’s core statements regarding treatment monitoring. A significant difference was found with regard to the evaluation of psychomotor development over the course of treatment; 49.3% of patients (group A, *n* = 302) and 24.8% of patients (group B, *n* = 327) (*p* < 0.001) were tested, respectively.

## 4. Discussion

Our multicenter study compared CH patient care documented in HypoDok with evidence-based recommendations. The majority of the surrogate parameters which assessed the adherence to the core statements were comparable in both groups, irrespective of guideline publication. However, a few significant differences between the two cohorts before and after guideline introduction were apparent. This persistent high level of care can be partly explained by the fact that the data of the registry is derived from pediatric endocrinologists, who were directly or indirectly involved in its development as members of the guideline-developing professional society. The present results were derived from centers of pediatric endocrinology representing only a minority of all CH patients in Germany and Austria and may therefore carry a selection bias.

HypoDok partly documented the recommendations regarding diagnostics (structural quality) as well as the implementation of therapy control (process quality). The low percentage of patients with documented TT4/fT4 serum concentration at initial diagnosis was one example of documentation gap, as low TT4/fT4 levels are a requirement for diagnostic confirmation. This documentation gap may arise due to a structural fault of the database, in that the date of diagnosis can be entered without underlying TT4/fT4 levels.

Another example was the low number of documented hearing tests at the start of treatment, although newborn hearing screening has been a standard service of public health insurance since 01/2009 in Germany and has been regularly reimbursed since 10/2010 [18]. This problem of incomplete documentation in HypoDok, especially at the beginning of treatment, is well known and has already been described by Ellerbroek et al. [14]. The reason for this may be that many patients were only referred to pediatric endocrinologists within the course of the first months, as the guideline only recommends referral within three months; retrospective documentation occurred in only 1/3 of these patients. This clearly demonstrates the importance of strengthening HypoDok’s awareness also outside the professional society to a wider spectrum of pediatricians combined with promoting and rewarding CH documentation, for example, through references within a revised guideline or short notes in journals. Currently, the registry is known mainly within the professional society, through presentations at the annual meetings and its presentation online which aims to target a larger audience [15].

Furthermore, improved documentation of psychomotor developmental assessment is required, especially in group B (24.8%; group A 49.3%). It may be assumed that these assessments were only carried out during the course of therapy if there was a clear clinical indication for developmental retardation. Experienced examiners preselect patients who need a formal assessment, which is, however, not according to current guideline recommendations. The rate of developmental assessments may be underestimated in group B, as some patients are still very young and therefore not all data have been collected so far. Further reasons why psychomotor development has not been regularly checked may be lacking compliance of the patient’s family. This may be due to fear of receiving an unfavorable result, a lack of understanding of the importance of the test, and the time required for the extensive testing procedure [14]. Here, the implementation of shorter screening tests could increase patient compliance.

In general, a very variable level of adherence to existing guidelines has already been observed by the treating physicians for a wide range of pediatric diseases. For example, guideline adherence regarding the use of antibiotics in infectious disease is a highly investigated subject [19,20,21]. These studies showed no change of management in urinary tract infections after introduction of the guideline, whereas the use of antibiotics in community-acquired pneumonia was evidence-based. Overall, it could be shown that non-compliance to evidence-based recommendations carries risks.

Vezzani et al. examined guideline adherence in adult patients with primary hypothyroidism [22]. Here, a satisfactory adherence to the European recommendations was found, with the exception of the L-T_4_ starting dose.

Clinical practice guidelines are important tools to practice evidence-based medicine. This was also concluded in a review using examples of pediatric endocrinologic disorders [23]. The successful integration of guideline recommendations can lead to a high, uniform standard of care, save financial resources, and improve patient outcome [24]: James et al. exemplified this through the study of the guideline’s use for the treatment of acute respiratory distress syndrome.

The aim to improve guideline implementation should also include the evaluation of patient compliance. Provision of practice aids like checklists or reminders for clinicians and implementation of diaries for patients may improve the problem next to a clear documentation of patient history by healthcare providers [25]. Matlock et al. followed this two-sided approach and presented methods to equally promote guideline adherence by improvement of patient care through the responsible physicians and also in a CH patient-based manner [26]: these included compiling and continuation of patient records in a database, and education programs of treating physicians and patient families on guideline recommendations. Furthermore, patient registries should also be strengthened as a suitable instrument for quality assurance [27].

The percentage of included patients in the HypoDok registry shows that the documented guideline adherence and the registry implementation do not strongly correlate, thus indicating a lack of documented adherence to the guidelines. This could be improved by specifically mentioning the existence of the registry in the guideline, which is currently being revised. At present, the registry is mainly known by pediatric endocrinologists through presentations during conferences of the professional society. However, it should also be promoted among general pediatricians (e.g., by reference in the guideline). Furthermore, a progressive digitalization of data collection and automated data transfer from the patient file to the registry would help to close the documentation gap.

As far as a deduction from documented adherence to actual patient adherence can be made, the HypoDok registry documented a satisfactory adherence to the core statements of the guideline regarding treatment. Thus, the basic preconditions for achieving the best possible patient outcome with adequate therapy were given in both cohorts [28,29,30]. The adherence concerning diagnostics and treatment monitoring by professionals, however, can be improved in both groups. Also, guideline recommendations on developmental assessment in routine care might require re-evaluation. The relationship between severity of illness and compliance was not assessed here. This would be an important aspect to be investigated in further studies.

Overall, the results cannot easily be transferred to the overall quality of care of patients with CH in Germany and Austria, as only 17.5% of expected patients with CH in Germany and even 10.3% in Austria are currently documented in HypoDok at the time period analyzed here. This results in a selection bias. This low reporting frequency is due to the fact that the database is known and used among experts (pediatric endocrinologists), but not widely used by pediatricians in private practice. This could indicate that a significant number of patients with CH are not seen by pediatric endocrinologists. The use of the HypoDok registry should be seen as an opportunity to document and ensure the quality of care for patients with CH and to adapt it to current clinical standards. This way, the treating physician can also compare and adapt patient care with the existing data on the basis of regular database reports.

So far only a few genetic alterations causing CH have been discovered and genetic testing in CH remains scarcely available in routine diagnostics (group A *n* = 11, group B *n* = 13). General implementation of genetic testing in CH in general patient care will increase over the course of the following years. Compiling of genetic sequencing data combined with clinical information and phenotypic data including blood tests could further help to subtype patients suffering from CH into clinically relevant subgroups.

Overall, management of patient data in databases has not only been identified by the professional society as one of the appropriate instruments for quality assurance. Administrative bodies such as the European Commission or the National Institute of Health also started programs for rare diseases [31]. One example is the European initiative of European Reference Networks (ERN) for different specialities like Endo-ERN for rare endocrine conditions in order to share knowledge to assure and maintain high level of patient care across Europe. As part of this initiative, patient cohort documentation in databases in the case of rare diseases such as CH is important to ensure structural quality [32,33]. In this context it would be additionally helpful to create a network of accredited regional centers for comprehensive CH management in childhood, closely linked to neonatal screening centers and primary care pediatricians. Furthermore, center-specific obligation for documentation will have to be strengthened, for example, by means of financial incentives. The authors acknowledge that the current setup of the registry (local documentation with periodic extraction of pseudonymized data) may not be optimal for pediatricians taking care of only few patients with CH. Long-lasting adherence to guidelines and respective documentation remains difficult, since in principle, it is easier to motivate participating professionals and patients over a limited period of time [34].

Finally, a legal requirement to keep these patients in registries should be discussed. As the low number of documented CH patients shows, the implementation of the guideline recommendations of the professional society can only be checked and improved by either obligation or standardized acquisition of data in order to guarantee adequate care. The fact that legal requirements and financial support by the state are helpful in this respect is not only reflected in the reporting frequency of newborn screening, but also in other fields of pediatrics. For example, in oncology, approximately 95% of childhood cancers are documented in the German Childhood Cancer Registry by appropriate legal requirements [35].

## 5. Conclusions

In summary, the here presented data suggest that in both groups, a guideline-adherent treatment initiation for CH is implemented. This is the prerequisite condition to achieve a regular physical and psychomotor development in patients with CH. However, direct documentation of psychomotor development seems difficult in routine care. HypoDok can currently only monitor a small proportion of affected patients, primarily those treated by pediatric endocrinologists. In addition, the assessment of guideline adherence based on the available data should be considered with caution because of the suggested documentation gap. In order to maintain the best possible standard of care, this rare disease should be treated in a center with expertise in pediatric endocrinology [9]. In order to monitor and further optimize care, more complete documentation in a registry designed for this purpose such as HypoDok is desirable. Here, support through development of law-enforced requirements in addition to a higher level of awareness of HypoDok may increase the frequency of documentation.

## Figures and Tables

**Figure 1 IJNS-07-00010-f001:**
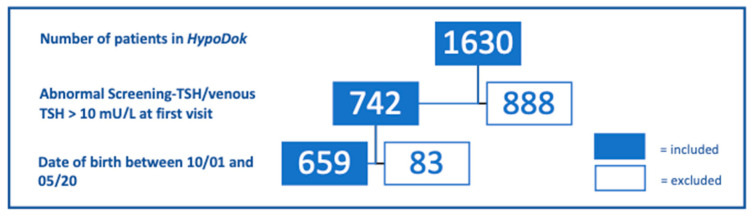
Eligible patients of the HypoDok registry, according to the inclusion criteria of this study (abnormal screening TSH/venous TSH > 10 mU/L).

**Table 1 IJNS-07-00010-t001:** Excerpt of statements included in the German Guideline for Primary Congenital Hypothyroidism (CH) [9].

Statement 1	Initial TSH determination sufficient to initiate confirmatory testing, TRH test not required
Statement 2	Confirmation of CH by low serum fT4 or TT4 concentrations Premature infants/neonates in intensive care: fT4 and TT4 are required at same time
Statement 3	Ultrasonography of thyroid gland should be performed Only restrictive use of scintigraphy
Statement 4	Hearing test: at confirmation and in the course of treatment
Statement 7	Start L-T_4_ therapy as early as possible At least before day 14
Statement 8	Initial daily L-thyroxine dose 10 µg/kg–15 µg/kg
Statement 9	Timepoints for follow-up: 1, 2, and 4 weeks after start of therapy Subsequently at 3-month intervals during the first two years of life
Statement 11	Monitoring of psychomotor development: During the first two years of therapy Before start of school

**Table 2 IJNS-07-00010-t002:** Demographic characteristics of CH patients, compared before and after guideline appearance (groups A and B).

	Group A “Before Guideline” 10/2001–1/2011	Group B “After Guideline” 2/2011–05/2020	*p*-Value
Gestational age (weeks)	40 (38; 41) (*n* = 314)	40 (38; 41) (*n* = 327)	1.00
Birth weight [g]	3370 (2951; 3690) (*n* = 320)	3395 (2970; 3730) (*n* = 330)	1.00
Birth length [cm]	51 (49; 53) (*n* = 259)	51 (49; 53) (*n* = 324)	1.00
10 min Apgar score	10 (8;10) (*n* = 322)	10 (8; 10) (*n* = 319)	1.00
Parental target height (Tanner) [cm]	172.5 (168.5; 176.5) (*n* = 271)	172.3 (168; 176) (*n* = 274)	1.00
Female [%]	63.6 (*n* = 327)	60.6 (*n* = 332)	1.00
Premature babies (<37 weeks) [%]	8.6 (*n* = 314)	10.4 (*n* = 327)	1.00

Results are given as median (Q1; Q3) or in %. Q1 = first quartile, Q3 = third quartile, Statistics: Wilcoxon Rank Sum Test/Chi-Square Test

**Table 3 IJNS-07-00010-t003:** Comparison of official number of CH newborn diagnoses per year in newborn screening and documentation in HypoDok for (a) Germany [3] and (b) Austria [17].

Year	Number of Confirmed Cases of CH in Newborn Screening	Number of New Cases of CH Documented in HypoDok	% N HypoDok/N Newborn Screening
(a) Germany			
2004	222	29	13.1
2005	187	22	11.8
2006	165	26	15.8
2007	163	35	21.5
2008	184	47	25.5
2009	195	52	26.7
2010	207	40	19.3
2011	207	36	17.4
2012	205	44	21.5
2013	211	43	20.4
2014	213	29	13.6
2015	235	29	12.3
2016	242	28	11.6
2017	279	39	14.0
average	208	36	17.5
(b) Austria			
2015	25	2	8.0
2016	32	2	6.3
2017	35	4	11.4
2018	24	1	4.2
2019	31	5	16.1
average	29	3	10.3

Results are given as absolute numbers and the HypoDok coverage is stated in percent.

**Table 4 IJNS-07-00010-t004:** Description and comparison of both groups according to the guideline recommendations: (a) diagnostics, (b) treatment, (c) treatment monitoring.

	Group A “Before Guideline” 10/2001–1/2011	Group B “After Guideline” 2/2011–05/2020	*p*-Value
(a) Diagnostics			
TT4 and/or fT4 determined at diagnosis [%] ^†^	34.9 * (*n* = 327)	24.4 * (*n* = 332)	0.02
Serum TSH determined at diagnosis [%] ^†^	36.1 * (*n* = 327)	23.2 * (*n* = 332)	0.002
Ultrasonography performed [%]	92.6 (*n* = 323)	92.1 (*n* = 328)	0.86
Scintigraphy performed [%]	5.0 (*n* = 318)	1.6 (*n* = 309)	0.17
Hearing test 14 days around start of treatment [%]	24.4 * (*n* = 172)	11.8 * (*n* = 229)	0.01
(b) Treatment			
Age at start of treatment [d]	6 (5; 9) (*n* = 316)	6 (5; 8) (*n* = 318)	0.55
Dose of L-T4 at start [µg/d]	50 (50; 50) (*n* = 122)	50 (50; 50) (*n* = 131)	1.00
(c) Treatment monitoring			
Follow-ups during first year of life [*n*]	5 (3; 7) (*n* = 293)	5 (3; 7) (*n* = 320)	0.58
Follow-ups during second year of life [*n*]	3 (2; 4) (*n* = 255)	3 (2; 4) (*n* = 259)	0.33
Follow-up 1 week after start of treatment [%]	24.2 (*n* = 327)	32.5 (*n* = 332)	0.17
Follow-up 2 weeks after start of treatment [%]	31.2 (*n* = 327)	38.0 (*n* = 332)	0.41
Follow-up 3–4 weeks after start of treatment [%]	34.0 (*n* = 327)	41.9 (*n* = 332)	0.29
Hearing test performed within the first 3 years of life (%)	46.4 (=276)	40.0 (*n* = 320)	0.58
Developmental assessment performed [%]	49.3 * (*n* = 302)	24.8 * (*n* = 327)	<0.001

Results are given as median (Q1; Q3) or in %. Q1 = first quartile, Q3 = third quartile, statistics: Wilcoxon Rank Sum Test/Chi-Square Test, * *p* < 0.01, ^†^ = even though “serum TSH determined at diagnosis” is part of the inclusion criteria for CH, the percentage of documented measurements is low as “abnormal screening TSH” is sufficient for patient’s inclusion in HypoDok. A documentation gap is to be assumed, since confirmatory testing is essential for diagnosis.

## Data Availability

The study was conducted according to the guidelines of the Declaration of Helsinki and was approved by the data protection commission of the Sachsen-Anhalt state review board and data collection for the registry was approved as previ-ously published [14].

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
