# Peer review of "Guideline Adherence and Registry Recruitment of Congenital Primary Hypothyroidism: Data from the German Registry for Congenital Hypothyroidism (HypoDok)"

_2409-515X, 2021, doi:10.3390/ijns7010010_

Round 1

Reviewer 1 Report

Although the group of patients with CH diagnosed with neonatal screening is relatively small (17.5% and 10.3%, respectively), the results indicate that differences in CH management were not significantly altered by the guidelines. The work would be enriched by the addition of data on CH screening, such as the screening incidence (recall rate) of CH, the real incidence of definitively diagnosed CH, the types of CH disorders (atyreosis, ectopia, dyshormonogenesis, etc.). This would complement the initial statement that "NS CH is mandatory, but medical care thereafter is inconsistent". The question for discussion is whether it would not improve the unification of procedures by creating a network of accredited regional centers for comprehensive CH management in childhood, closely linked to neonatal screening centers and primary care pediatricians.

Author Response

Point 1: The work would be enriched by the addition of data on CH screening, such as the screening incidence (recall rate) of CH, the real incidence of definitively diagnosed CH, the types of CH disorders (atyreosis, ectopia, dyshormonogenesis, etc.). This would complement the initial statement that "NS CH is mandatory, but medical care thereafter is inconsistent".

Response 1: We thank the reviewer for the opportunity to improve our work. In an additional table (Table S2) we provide additional information of the newborn screening report of the German Society for Newborn Screening (DGNS) including the suggested measures of recall rate of CH, real incidence of definitively diagnosed CH, the types of CH disorders. Unfortunately, these data were not available for Austria.

Furthermore, we have provided information (Table S3) including a sonography-based morphological classification of the registry HypoDok, which allows a subclassification of patients with athyreosis, ectopic thyroid gland and thyroid gland in loco typico. The available data are presented subdivided into the two groups A (before guideline introduction) and B (after guideline introduction). In addition, we show the available data of genetic examinations in Table S4. Findings here are divided in mutations which causes a) dysgenesis and b) dyshormonogenesis.

We hope that we sufficiently addressed the reviewers point by the inclusion of new data regarding the screening incidence (page 4 line 144-148 + Table S2), the morphology of thyroid gland (page 5 line 163-167, + Table S3) and genetic findings (page 5 line 167-170 + Table S4).

Point 2: The question for discussion is whether it would not improve the unification of procedures by creating a network of accredited regional centers for comprehensive CH management in childhood, closely linked to neonatal screening centers and primary care pediatricians.

Response 2: We thank the reviewer for this helpful suggestion to further enrich the discussion part of our manuscript by including the important point, that the network of regional centers for comprehensive CH management in childhood, neonatal screening centers and primary care pediatricians will be one important measure to improve standardized operational procedures in CH. This could be implemented in already existing structures, such as EndoERN.

We have now included this point into the discussion section (page 9, line 300-302).  

Reviewer 2 Report

The article presents data from the German registry for the follow up of congenital hypothyroidism detected by neonatal screening, including patients from Germany born Oct 2001-May 2020 and Austria 2015-2019. Registered were in average 17,5% and 10,3% of screening detected CH patients in the respective countries. Guidelines for the follow up of the patients were published in Feb 2011 and comparisons were made between the frequency of registration of German patients in the registry before and after the publication of the guideleines and a small decrease was observed. The authors plead for resources to increase the use of the registry with the aim to improve treatment and outcome. It would have been interesting to learn what measures were taken to improve the frequency of registration in the registry in addition to the implementation of the guidelines.

Author Response

Point 1: It would have been interesting to learn what measures were taken to improve the frequency of registration in the registry in addition to the implementation of the guidelines.

Response 1: This is an important point taken up by the reviewer and we thank for the opportunity to further clarify this aspect.

We fully agree that continuous efforts are necessary to increase the awareness of the registry to make its broad implementation in patient care feasible. So far, to spread the knowledge about the registry within the professional society to increase documentation rate, the registry has been introduced to the professional society on multiple occasions through oral presentations during the annual meetings of the professional society of Paediatric Endocrinology in Germany.

To make the information openly accessible to a large audience, a freely accessible website by the German Society for Paediatric Endocrinology and Diabetology (https://buster.zibmt.uni-ulm.de/projekte/Hypothyreose/) was created. This information source allows collaborating centres to get an informed current update about the registry and aims to recruit new participating centres).

We have now included this point into the discussion section (page 7, line 214-217) in addition to the existing mention at page 8, line 260-261.